# End-to-End on-device Federated Learning: A case study

## Abstract

With the development of computation capability in devices, companies are eager to utilize ML/DL methods to improve their service quality. However, with traditional Machine Learning approaches, companies need to build up a powerful data center to collect data and perform centralized model training, which turns out to be expensive and inefficient. Federated Learning has been introduced to solve this challenge. Because of its characteristics such as model-only exchange and parallel training, the technique can not only preserve user data privacy but also accelerate model training speed. In this paper, we introduce an approach to end-to-end on-device Machine Learning by utilizing Federated Learning. We validate our approach with an important industrial use case, the wheel steering angle prediction in the field of autonomous driving. Our results show that Federated Learning can significantly improve the quality of local edge models and reach the same accuracy level as compared to the traditional centralized Machine Learning approach without its negative effects. Furthermore, Federated Learning can accelerate model training speed and reduce the communication overhead, which proves that this approach has great strength when deploying ML/DL components to real-world embedded systems.

## 1 Introduction

With the development of computation capability in devices, Machine Learning and Deep Learning arouse great interests by companies who are eager to utilize ML/DL methods to improve their service quality. However, with the explosive growth of data generated on edge devices, the traditional centralized Machine Learning approaches have shown its weakness, such as data communication overhead, model compatibility, training efficiency, etc. (L'heureux et al., 2017a) Figure 1 illustrate a traditional Machine Learning approach with the centralized learning framework.

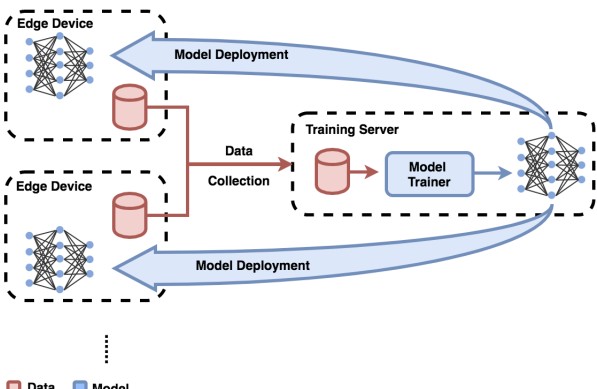

Figure 1: Traditional Centralized Learning System

The diagram contains four stages: 1) data collection from multiple distributed edge devices 2) model training in a central server 3) model validation based on existing testing data 4) model deployment

to edge devices. However, the data collected from edge devices need to be transmitted to a central server and perform model training on that enormous data set, which turns out to be inefficient and expensive. In order to solve these challenges, Federated Learning has been introduced as an efficient approach which can distribute learning tasks to the edge devices and avoid massive data transmission. Furthermore, due to the characteristics of Federated Learning, on-device training becomes possible and the local model quality can be continuously improved.

Although the concept of Federated Learning has significant benefits and potential in AI engineering fields, it is hard for industries and companies to build a reliable and applicable on-device Federated Learning system. Some previous research identified the challenges of deploying AI/ML components into a real-world industrial context. As defined in *"Engineering AI Systems: A Research Agenda"* (Bosch et al., 2020), AI engineering refers to AI/ML-driven software development and deployment in production contexts. We found that **the transition from prototype to the production-quality deployment of ML models proves to be challenging for many companies** (L'heureux et al., 2017b) (Lwakatare et al., 2019).

The contribution of this paper is threefold. First, we utilize Federated Learning, a distributed machine learning technique, and validate it on an important industrial use case, steering wheel prediction in the field of autonomous driving, which is also a classic end-to-end learning problem. Second, we describe an end-to-end on-device Federated Learning approach to efficiently train Machine Learning models in a distributed context. Third, we empirically evaluate our approach on the real-world autonomous driving data sets. Based on our results, we demonstrate the strength of Federated Learning compared to traditional centralized learning methods.

The remainder of this paper is structured as follows. Section 2 we introduce the background of this study. Section 3 details our research method, including the simulation testbed, the utilized machine learning method and the evaluation metrics. Section 4 presents the end-to-end Federated Learning approach utilized in this paper. Sections 5 evaluates proposed learning approach to empirical data sets. Section 6 outlines the discussion on our observed results. Finally, Section 7 presents conclusions and future work.

## 2 BACKGROUND

The first Federated Learning framework was proposed by Google in 2016 Konečnỳ et al. (2016), The major objective of Federated Learning is to learn a global statistical model from numerous edge devices. Particularly, the problem is to minimize the following finite-sum objective function 1:

$$\min_w f(w), \ \ where \ f(w) := \sum_{i=1}^{n} \lambda_i f_i(w) \tag{1}$$

Here, $w$ represents model parameters, $n$ is the total number of edge devices, and $f_i(w)$ is the local objective function which is defined by high dimensional tensor $w$ of the $i_th$ device. $\lambda_i$ ($\lambda_i \geq 0$ and $\sum_i \lambda_i = 1$) gives the impact of $i_th$ remote device and is defined by users. This formula is also applied throughout this research.

With the development of the concept of cloud computing and decentralized data storage, there has been increasing interest in how to utilize this technique to improve Machine Learning procedure. There are two classic applications which were realized by Hard et al. (2018) and Ramaswamy et al. (2019). Authors applied Federated Learning techniques on the Google Keyboard platform to improve virtual keyboard search suggestion quality and emoji prediction. Their results show feasibility and benefit of applying federated learning to train models while preventing to transfer user's data. However, authors in previous research didn't discuss the impact of model training time and the communication cost when deploying and training models on edge devices. Furthermore, due to the system environment and troubles encountered when deploying Federated Learning into different cases, we propose an end-to-end approach and validate the on-device Federated Learning into a completely different industrial scenario, the steering wheel angle prediction.

With the inspiration of the work by Bojarski et al. (2016), we designed and developed a deep convolutional neural network to directly predict the steering wheel angle and control the steer based on the prediction. The training data is collected from single images sampled from video and the

ground truth is recorded directly from real-time human behavior. In order to improve the model prediction performance, a two-stream model was first proposed in Simonyan & Zisserman (2014) and applied in Fernandez (2018) due to its robustness and lower training cost compared with other networks such as 3D-CNN (Du et al., 2019), RNN (Eraqi et al., 2017) and LSTM (Valiente et al., 2019). However, the previous research for this use case is mainly focusing on training model in a single vehicle. In this paper, we will apply Federated Learning to accelerate model training speed and improve the model quality by forming a global knowledge of all participating edge vehicles.

## 3 METHOD

In this research, the empirical method and learning procedure described in Zhang & Tsai (2003) was applied to make a quantitative measurement and comparison between Federated Learning and traditional centralized learning methods. In the following sections, we present the mathematical notations used in this paper, our testbed, data traces and the convolutional neural network architecture utilized for solving the problem of steering wheel angle prediction.

### 3.1 MATHEMATICAL NOTATIONS

We first introduce the mathematical notations that will be used in the rest of the paper:

$\boldsymbol{A}_t$          An image frame matrix at time $t$

$\boldsymbol{O}_t = f(\boldsymbol{A}_t, \boldsymbol{A}_{t-1})$          An optical-flow matrix at time $t$

$\theta_t$          Steering wheel angle at time $t$

### 3.2 DATA TRACES AND TESTBED

The datasets used in this paper is SullyChen collection of labeled car driving data sets, which is available on Github (SullyChen, 2018). In this collection, there are two datasets (Dataset 2017 and Dataset 2018) which record different driving information on different routes.

Dataset 2017 contains approximately 45,500 images, 2.2 GB. The dataset records a trajectory of approximately 4km around the Rolling Hills in LA, USA in 2017. 2017 dataset is used for pretraining the model. (The model will be used to initialize edge models before Federated Learning)

Dataset 2018 contains approximately 63,000 images, 3.1 GB. This dataset records a trajectory of approximately 6km along the Palos Verdes in LA. 2018 dataset is used for end-to-end Federated Learning and model validation. In order to provide fruitful evaluation, we conducted experiment on 4, 8, 16, 32 and 64 edge vehicles. The data were divided into corresponding number of parts and distributed to edge vehicles. Besides, in each edge vehicle, 70% of local dataset were training set while 30% were acted as the testing set.

In each edge vehicle, the first 70% data are regarded as the previously recorded driving information while the rest 30% are future information. The models were continuously trained based on the recorded information and perform prediction and validation on the steering wheel angle information by using future driving data.

Table 1 provides the hardware information for all of the servers. In order to simulate aggregation and edge functions, one server is adopted as the aggregation server while the rest are acted as edge vehicles.

### 3.3 MACHINE LEARNING METHOD

A two-stream deep Convolutional Neural Network (CNN) (Simonyan & Zisserman, 2014) (Fernandez, 2018) is utilized to perform angle prediction. Figure 2 gives the detailed information about the architecture. In our implementation, each stream has two convolutional layers and a max-pooling layer. After concatenating, there are two fully-connected layers which are activated by ReLU function.

Table 1: Hardware setup for testbed servers

| CPU | Intel(R) Xeon(R) Gold 6226R |
| --- | --- |
| Cores | 8 |
| Frequency | 2.90 GHz |
| Memory | 32 GB |
| OS | Linux 4.15.0-106-generic |
| GPU | Nvidia Tesla T4 GPU (Only in edge vehicles) |

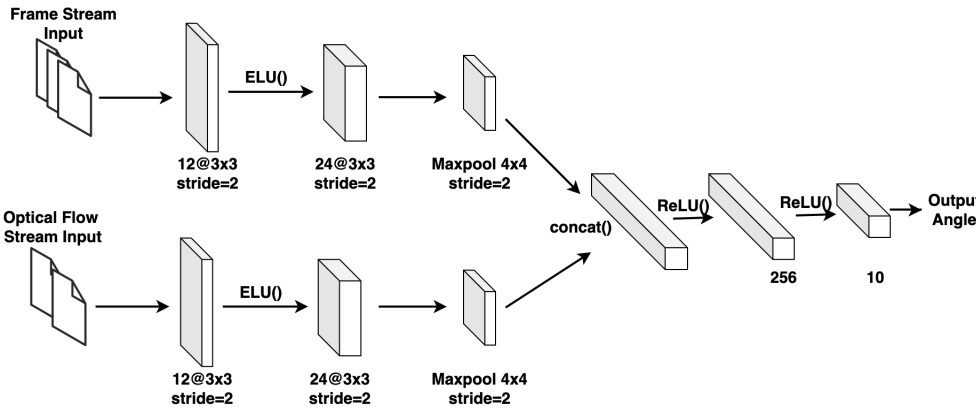

Figure 2: Model description

The model contains two different neural branches which consume spatial information and temporal information as the inputs of two streams and then output the predicted steering angle. For the first stream the model consumes 3 frames of RGB images, which can be denoted as $\{A_{t-2}, A_{t-1}, A_t\}$. The second stream is the two-frame optical flow calculated by two consecutive frames $O_{t-1} = f(\{A_{t-2}, A_{t-1}\})$ and $O_t = f(\{A_{t-1}, A_t\})$.

Optical flow is a common temporal representation in video streams, which captures the motion changes between two frames (Horn & Schunck, 1981). The method of calculating optical flow applied in this paper is based on Gunnar Farneback's algorithm implemented in OpenCV (Farnebäck, 2003). Figure 3 demonstrate an example optical flow matrix produced by two consecutive image frame.

The process of training an local CNN network is to find the best model parameters which cause the minimum difference between the predicted angle and the ground truth steering angle. Therefore, in this case, we choose mean square error as the local model training loss function:

$$Loss = \frac{1}{N} \sum_{t=1}^{N} (\theta_t - \hat{\theta}_t)^2 \tag{2}$$

Here, $N$ represents the batch size while $\theta_t$ and $\hat{\theta}_t$ represent the ground truth and the predicted steering wheel angle value at time $t$.

During the process of model training in each edge vehicles, all the image frames will be firstly normalized to $[-1, 1]$. The batch size is 16 while the learning rate is set to $1e-5$. The optimizer utilized here is Adam (Kingma & Ba, 2014), with parameters $\beta_1 = 0.6$, $\beta_2 = 0.99$ and $\epsilon = 1e-8$.

### 3.4 EVALUATION METRICS AND BASELINE MODEL

In order to provide fruitful results and evaluation, we selected three metrics and two baseline models. The three metrics includes angle prediction performance, model training time and bandwidth cost:

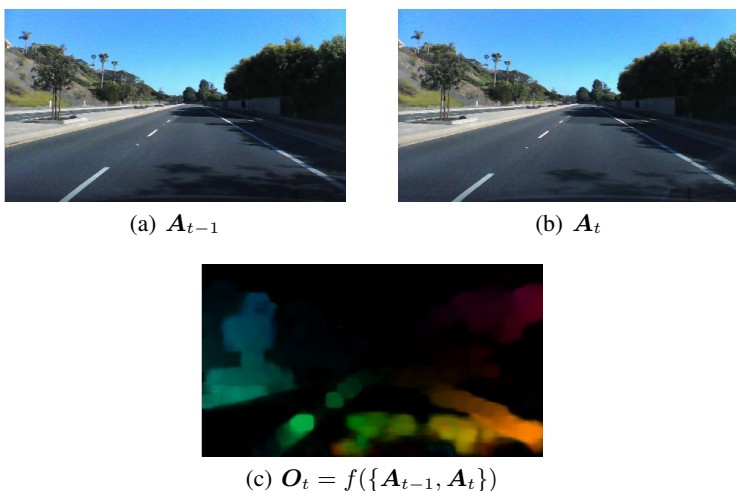

(a) $\boldsymbol{A}_{t-1}$                    (b) $\boldsymbol{A}_t$

(c) $\boldsymbol{O}_t = f(\{\boldsymbol{A}_{t-1}, \boldsymbol{A}_t\})$

Figure 3: Example of the optical flow

- **Angle prediction performance**: We use root mean square error (RMSE), a common metric, to measure the difference between prediction results and ground truth. The metrics can provide good estimation about the quality of trained model in each edge vehicles.

- **Model training time**: This metric is defined as the time cost for training a model at the edge vehicles. The result is the average of four edge vehicles during one training round. This metric demonstrates the speed of local edge devices updating their knowledge which is crucial and important for those systems which need to quickly evolve to adapt to the rapidly-changed environment. The metrics was measured in all the vehicles by checking model deployment timestamp.

- **Bandwidth cost**: This metric is defined as the total number of bytes transmitted during the whole training procedure. This metric demonstrate the total communication resources cost of achieving an applicable CNN model.

The two baseline models includes model trained by applying traditional centralized learning approach and the locally trained model without model sharing:

- **Traditional Centralized Learning model (ML)**: This baseline model is trained under the traditional centralized learning approach. Before model training, all the data from edge vehicles are firstly collected to a single server. The hyper-parameter of this model training is the same as Federated Learning which is mentioned in section 3.3. The performance can be then compared with the model trained by Federated Learning approach.

- **Locally trained model without model sharing (Local ML)**:

  This baseline models are trained directly on each edge vehicles. However, different from Federated Learning, there will be no model exchange during the training procedure. The prediction performance can be compared with Federated Learning model to see how Federated Learning can outperform those independently trained local models.

## 4 END-TO-END FEDERATED LEARNING

In this section, we describe the algorithm and the approach applied in this paper. In order to perform on-device end-to-end learning based on the input image frames, images are firstly stored in an external storage driver located on each edge vehicles. At the same time, the optical flow information are calculated. When triggering the training threshold, image frames and optical flow frames are fed into a convolutional neural network. The output of the network is compared to the ground truth for that image frame, which is the recorded steering wheel angle. The weights of the CNN are adjusted using back propagation to enforce the model output as close as possible to the desired output. Figure 4 illustrate the diagram of the learning procedure in a single edge vehicle.

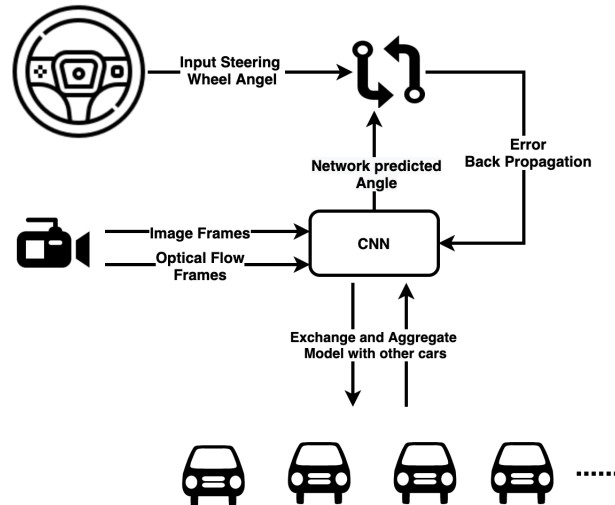

Figure 4: Diagram of end-to-end on-device learning procedure in a single vehicle

After finishing each training epoch, models in edge vehicles will also be updated to the aggregation server and form a global knowledge among other cars (Figure 5). The aggregation applied in this paper is FedAvg (Li et al., 2019), which is a commonly used Federated Learning algorithm in most of the research. The steps of FedAvg algorithm is listed below:

Step 1: Edge vehicles locally compute the model; After finishing each five local training epoch, they send updated model results to the aggregation server.

Step 2: The central server performs aggregation by averaging all updated models to form a global knowledge of all local models.

Step 3: The aggregation server sends back the aggregated result to each edge vehicles.

Step 4: Edge vehicles replace the local model and performs further local training by using the global deployed model.

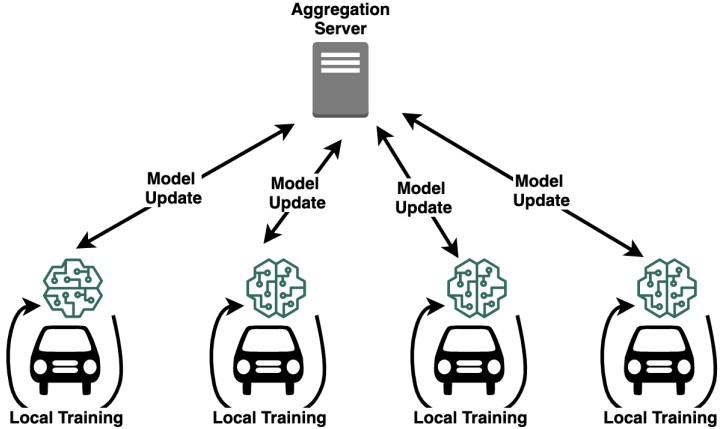

Figure 5: Process of Federated Learning

## 5 EVALUATION

In this section, we present the experiment results of the presented end-to-end on-device Federated Learning approach on the use case of steering wheel angle prediction. We evaluate the system performance in three aspects (The metrics are defined in section 3.4) - (1) Angle prediction performance (2) Model Training Time (3) Bandwidth cost.

Figure 6 illustrate the angle prediction performance between the model trained by Federated Learning (FL) and the locally trained model without any model exchange (Local ML). The results demon-

strate that the performance of traditional centralized trained model behaves similar to Federated Learning model. Besides, compared with independently trained model, Federated Learning can provide better prediction which is much closer to the ground truth.

Numeric results are provided in Table 2. We show detailed results with 4 vehicles participated in Federated Learning, which provides a clear view of prediction performance in each edge vehicle. The results illustrate that in vehicle 1 and 4, model of Federated Learning outperform other baseline models. In vehicle 2 and 3, model of Federated Learning only perform about $1^{\circ}$ worse than the traditional centralized learning model. Based on our results, we can summarize that Federated Learning model can provide more accurate prediction than local independently trained model and the behaviour of Federated Learning model can reach the same accuracy level compared with centralized learning model.

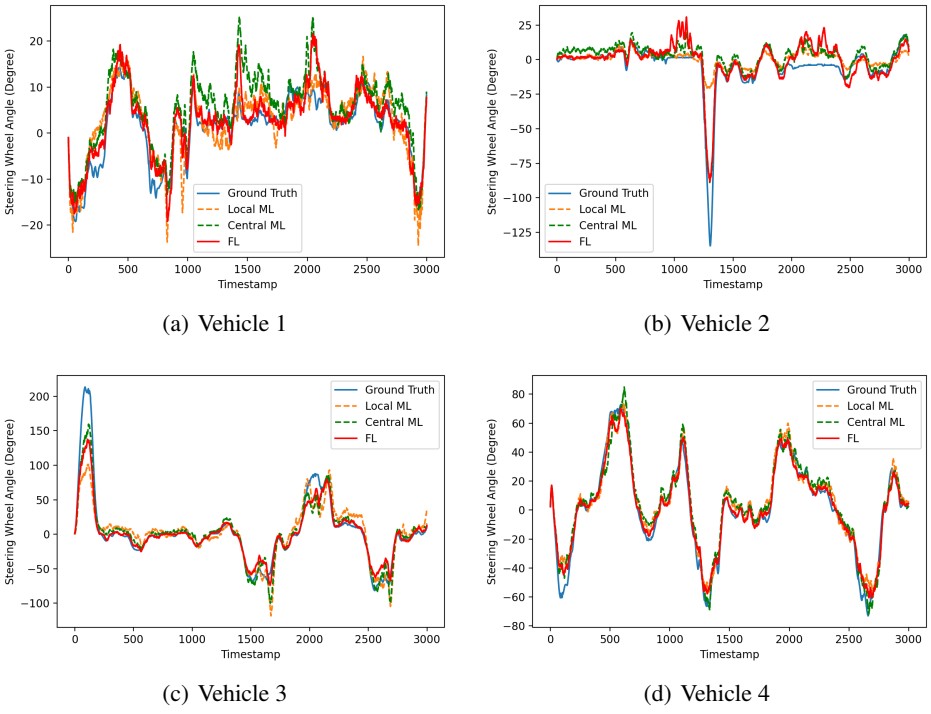

Figure 6: The comparison of angle prediction performance on four local vehicle test set with Federated Learning and two baseline models

|  | Vehicle 1 | Vehicle 2 | Vehicle 3 | Vehicle 4 | Overall |
|---|---|---|---|---|---|
| FL | 3.154 | 8.875 | 17.209 | 5.581 | 10.242 |
| ML | 5.371 | 7.914 | 16.215 | 7.258 | 10.099 |
| Local ML | 4.017 | 14.775 | 25.670 | 7.313 | 15.419 |

Table 2: Steering wheel angle regression error (RMSE) on test set of each edge vehicle (4 vehicles in total)

Table 3 gives the comparison of total training time and bytes transferred between Federated Learning and two baseline model. The total number of training epochs for all the models is 100 and the model training is accelerated by Nvidia Tesla T4 GPU. The results show that Federated Learning need slightly more training time than independently locally trained model due to the model exchange time cost. However, the training time of Federated Learning is reduced about 75% and we save about 25% bandwidth compared with traditional centralized learning method.

In order to evaluate the impact of different number of learning vehicles, we perform more experiments with 8, 16, 32, 64 vehicles participated. Table 4 gives the overall steering angle prediction

|                               | FL    | ML     | Local ML |
|-------------------------------|-------|--------|----------|
| Total Training Time (sec)     | 511.6 | 2137.2 | 485.3    |
| Total Bytes Transferred (GB)  | 1.56  | 2.02   | -        |

Table 3: Training Time and Bandwidth cost with different model training methods (4 Vehicles in total)

error and total training time of Federated Learning model with different number of vehicles. The overall value provides an overview of prediction performance among all of the test datasets belongs to all vehicles. With the increasing number of edge vehicles, the model prediction performance on the edge is further enhanced. Furthermore, total model training time is linearly decreased corresponding to the increasing number of edge vehicles. Based on our results, we can summarize that with the participation of more edge vehicles and the larger size of the input datasets, the advantages of Federated Learning will become more obvious.

| Number of Vehicles         | 4      | 8      | 16    | 32    | 64    |
|----------------------------|--------|--------|-------|-------|-------|
| Error (RMSE)               | 10.242 | 10.649 | 9.644 | 9.387 | 9.251 |
| Total Training Time (sec)  | 511.6  | 264.3  | 125.1 | 65.3  | 31.7  |

Table 4: Overall steering wheel angle regression error (RMSE) and model training time of Federated Learning model with different number of vehicles participated

## 6 DISCUSSION

Based on our experiment results, end-to-end on-device Federated Learning approach has more advantages compared with commonly used centralized learning approach. Federated Learning model can achieve same level of model prediction accuracy but decrease model training time and the bandwidth cost. Furthermore, if we compared with independently local trained model, because of the model sharing mechanism, Federated Learning can form a global knowledge of the whole datasets which are belongs to different participated edge vehicles. The model quality are largely enhanced and can achieve much better results.

Due to those advantages, there are a variety of other meaningful use cases that end-to-end on-device Federated Learning can help. The technique reported in this paper can not only be used for steering angle prediction in self-driving vehicles but also other on-device applications, such as camera sensors and motion detection, which requires continuously machine learning model training on the resource-constrained edges. Furthermore, because of the user data privacy and network bandwidth constraints, Federated Learning can be applied in those systems which need quickly-evolved model to adapt their rapidly changing environment.

## 7 CONCLUSION

In this paper, we describe an approach to end-to-end on-device Machine Learning by utilizing Federated Learning. We validate our approach with the wheel steering angle prediction in the self-driving vehicles. Our results demonstrate the strength and advantage of the model trained under end-to-end Federated Learning approach. The model can achieve the same level of prediction accuracy compared with commonly used centralized learning method but reduces training time with 75% and bandwidth cost with 25 % in our case. Note that if the number of participating devices is further increased, the reduction will be more obvious and the strength of Federated Learning will become stronger.

In the future we plan to validate our approach in more use cases. Also, we would like to explore more advanced neural network combined with Federated Learning method. Furthermore, we plan to find more suitable aggregation algorithms and protocols for our end-to-end on-device Federated Learning approach.

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
