# OpenReview forum: "End-to-End on-device Federated Learning: A case study"
_ICLR.cc/2021/Conference — Reject_

### Official Review · AnonReviewer2 · 2020-10-25
**Evaluation of on device federated learning for steering wheel angle prediction**

**Rating:** 6
**Confidence:** 4

**Review:**

The study evaluates federated learning (FL) in the context of steering wheel angle prediction, which is relevant for autonomous driving systems. Authors compare against two baselines a centrally-computed and locally-computed models and measure prediction error, training time and bandwidth cost. The work evaluates an existing approach and therefore its novelty and impact is limited. It does provide an interesting evaluation of FL for a relevant use case. Federated learning, as the authors indicate in the manuscript, is a promising approach for training ML applications while preserving user privacy, which is key to many industrial ML applications such as voice assistants and computer vision algorithms. For that reason, the impact of the paper is significant despite not being very original. The authors carry out a very simple study, but which seems sufficient to demonstrate that FL can have computational advantages, namely reduced training times and bandwidth costs. A challenging application of FL are ML applications that run on small devices that people carry around all the time, such as mobile phones and wearable devices.  In that scenario, there is the additional constraint that resources for training models on device are typically limited,  the smaller the device the more limited. An interesting extension of this study would be evaluate amount of computational resources used on the device as an additional evaluation metric. It would be great if the authors could add this metric to the present paper, but it could also be something for a followup publication, in other words, I do not think is needed for this paper to be published.

[Update after author's rebuttal]
I do not see any reason to modify my rating. I also identified the self-citation , but it did not affect my rating or evaluation of the paper.

---

### Official Review · AnonReviewer4 · 2020-10-25
**Nice case study for Federated Learning on autonomous driving application but no actual research proposed.**

**Rating:** 4
**Confidence:** 4

**Review:**

This paper presents a case study that applies Federated Learning for steering angle prediction in self-driving cars. All methods used have been previously proposed in the literature.

Pros:
+ A case study for an industrial use of federated learning (in autonomous driving application).
+ Results do show that Federated Learning can give accuracy close to a centralized model for this application but without having to send data to the server (thus saving training time and communication bandwidth requirements).

Cons:
- No actual research contribution since nothing new is proposed in this paper.
- While the training time and communication bandwidth savings are a good validation, this is not surprising since Federated Learning has been shown to have this benefit for many applications.

========== UPDATE AFTER REBUTTAL ===========
I have read the author's response. While the case study for industrial applications is important, it would probably be much more impactful if the same study was done on a much larger/realistic scale. For instance, right now it appears that each edge vehicle gets an already available dataset for federated learning, which may have been cleaned and preprocessed properly. For claiming a real industrial deployment/importance, it would have been great if the study was conducted with vehicles receiving real-time data from real vehicles which is prone to be extremely noisy (although the reviewer is not sure if this would be possible for regulatory reasons (e.g., if such learning experiments would be safe enough on real autonomous vehicles as these applications are safety-critical)). Currently, the paper neither has significant enough contributions from novelty side, nor from industrial deployment angle. Hence, as such, the paper cannot be accepted. Perhaps more application-oriented conferences maybe more suitable for this kind of work.

---

### Official Review · AnonReviewer3 · 2020-10-28
**An implementation of federated learning on a use case in autonomous driving. *The paper is not properly anonymized***

**Rating:** 2
**Confidence:** 5

**Review:**

******************************************************************************

The paper is not properly anonymized. The intro refers to “Our previous research” and says “As we defined in Engineering AI Systems: A Research Agenda, (Bosch et al., 2020), … .” As such it violates the anonymity policy.
******************************************************************************

This paper describes end-to-end implementation of Federated Learning (FL) on a use case of steering wheel prediction in autonomous driving. It provides empirical evaluation on real-world autonomous driving datasets and shows improved performance compared to centralized learning methods.

Pros:
Is it interesting to see an implementation of FL on a real-world use-case. The paper also does well in comparing different factors such as training time and bandwidth cost for FL and centralized training.

Cons:
The paper doesn’t have enough technical depth to be accepted at ICLR and reads more like a report than a paper. It mainly describes the implementation of FL for a real-world application, which, although important, does not contribute to the field in terms of developing better algorithms or better understanding the current ones.

A large part of the experiment section describes the hardware features, network structure and training method in great details, which seems redundant or unnecessary for an ICLR submission. For example, section 4 reads “The weights of the CNN are adjusted using back propagation to enforce the model output as close as possible to the desired output.”, which is obvious to most readers.

There are also some statements in paper that are not quite scientific or concrete. For example, the intro reads “due to the characteristics of Federated Learning, on-device training becomes possible.” This is not true as on-device training is not becoming possible due to FL, though FL certainly requires it.

---

### Official Review · AnonReviewer1 · 2020-10-29
**maybe consider other more application venues?**

**Rating:** 4
**Confidence:** 4

**Review:**

This paper applies federated learning to  steering wheel prediction for autonomous driving. "Federated learning" in this draft mainly refers to an on-device distributed training algorithm where each edge device hosts its private data and performs local updates (model training) and send the updates back to a central server to aggregate. More specifically, this paper uses the most well-known algorithm in federated learning, FedAvg (McMahan et al. 2017).

Pros
+ The application is real and seems important.
+ Distributed/federated learning makes sense for this application.

Cons
- The main contributions of the draft are not clear. It looks to me such empirical studies of a well-known algorithm on a specific application will better fit a more application-oriented or system-oriented venue, e.g., CVPR, SysML.
- How are the hyperparameters tuned for centralized and federated setting?
- What are the hardwares on edge devices/vehicles, and what are the hardwares in datacenter for centralized training? The draft mentioned Tesla T4 GPUs, but it seems not clear exactly how much computation power has been used.
- Could the authors clarify "companies need to build up a powerful data center to collect data and perform centralized model training, which turns out to be expensive and inefficient. Federated Learning has been introduced to solve this challenge."? As far as I know, the primary motivation for federated learning is privacy protection. Edge devices has far less computation power and big communication barrier, why would it solve "this challenge"?
- The following sentences seem to against the anonymous rules? "Our previous research shows the challenges of deploying AI/ML components into a real-world industrial context. As we defined in ”Engineering AI Systems: A Research Agenda” (Bosch et al., 2020), AI engineering refers to AI/ML-driven software development and deployment in production contexts. We found that the transition from prototype to the production-quality deployment of ML models proves to be challenging for many companies (L’heureux et al., 2017b) (Lwakatare et al., 2019)."

Some minor improvement:
The abbreviation “ML/DL” seems never introduced
It seems unnecessary to capitalized “Machine Learning”, “Federated Learning”.
Consider cite the original FedAvg (McMahan et al. 2017) paper instead of (Li et al., 2019).

====== post rebuttal ======

I do not think the response addressed my concerns. I would strongly suggest authors reconsider the design choices where I raised questions. Note that these are not only clarification questions, but also fundamentals of machine learning and federated learning.

---

### Decision · Program_Chairs · 2021-01-07
**Final Decision**

**Decision:**

Reject

**Comment:**

This paper proposes the use of federated learning to the application of steering wheel prediction for autonomous driving. While the application is new and interesting, the reviewers felt that the approach and results were mostly empirical. I suggest that the authors improve the conceptual/algorithmic contribution of the paper in a revised draft. Another suggestion is to include a better explanation of hyper-parameter optimization used in the experiments. I hope that the reviewers' constructive comments will help the authors revise the draft adequately for submission to a future venue!